# The Impact of COVID-19 on Motivation, Involvement, and Behavior of Cyclists in Taiwan

**DOI:** 10.3390/bs12120479

**Published:** 2022-11-25

**Authors:** Ya-Ling Yu, Jia-Yi Lin, Chiung-Hsia Wang, Chin-Huang Huang

**Affiliations:** 1Department of Sport Management, National Taiwan University of Sport, Taichung 404, Taiwan; 2Department of Physical Education, National Taiwan University of Sport, Taichung 404, Taiwan

**Keywords:** COVID-19, motivation, involvement, cycling tourism, behavior intention, wellbeing

## Abstract

Coronavirus disease (COVID-19) has spread all over the world and has impacted tourism globally, with countries taking various measures such as travel restrictions, border closures, lockdowns, or quarantines to contain the virus. Tourists’ motivation has also been affected by COVID-19, but so far, the literature has not yet discussed their concern over COVID-19 as well as the relationships among their motivation, involvement, and behavior intention. Therefore, this study fills the gap in the literature by taking cycling tourism as an example to understand the involvement of tourists concerning COVID-19 and presents the depth and breadth of its effects upon tourism. Due to the challenge of face-to-face, on-site investigation, we employ an online survey for data collection, use exploratory factor analysis to extract the main factors of motivation, involvement, and behavior intention, and set up a structural equation model to examine the relationships among the three factors. The results show that COVID-19 has positively and significantly affected motivation and involvement. Motivation positively and significantly affects involvement, and involvement affects motivation and behavior intention. The main finding herein is that motivation does not affect behavior, but involvement does mediate between the motivation and behavior of cyclists during COVID-19. Therefore, people may perceive the risk of health and wellbeing through such involvement.

## 1. Introduction

The COVID-19 pandemic has become a barrier for mobility throughout the global tourism system. Many countries set up travel restrictions, border closures, lockdowns, or quarantines to control the leisure and recreation behaviors of people. For most countries’ citizens, the main alternative has shifted from foreign destinations to domestic tourism with sites close to home [1,2]. According to the Committee for the Coordination of Statistical Activities [3], international tourist arrivals have fallen by 74%, from almost 1.5 billion arrivals in 2019 to 381 million in 2020. During this period, international arrivals dropped 84% in Asia-Pacific, while the demand for domestic tourism in most nations has increased rapidly.

The restriction on tourism has not only reduced the number of foreign tourists, but has also negatively impacted many economies. To mitigate the effect, many countries tried to reopen their border after vaccines became available. Indeed, the COVID-19 pandemic has been a crucial factor in changing administrations’ and individuals’ decision making regarding tourism. Motivation is one main factor driving individuals to visit a destination and to participate in recreation activities [4] and is also an antecedent of enduring involvement [5,6,7], which is reflected by people devoting themselves to an activity or associated product [8,9]. Many COVID-19 studies have focused on motivation [10], motivation and intention [11], motivation and constraint [12], or behavior and intention [13,14]. Thus far, no study has discussed tourists’ concern over COVID-19 as well as the relationships among their motivation, involvement, and behavior intention. This study is the first to present the depth and breadth of COVID-19 influences upon cycling tourism.

The purpose of this study is to examine how the COVID-19 pandemic impacts the motivation and behavior of tourists and whether involvement mediates between the motivation and behaviors of tourists by adopting cycling tourism data in Taiwan. Taiwan’s bicycle industry is famous worldwide, and cycling is the country’s second-most popular exercise. Before the pandemic, in 2017, there were 5.1 million cyclists in Taiwan [15]. To understand the COVID-19 impacts on cycling tourism, the research first examines the motivation of cyclists and then analyzes the involvement of cyclists via their inherent motives, needs, and interests in regard to COVID-19. Finally, it explores the response behavior of cyclists to motivation and involvement under the pandemic. Interdisciplinary research, such as neuroimaging techniques, can be applied in social science [16], but to prevent the infection of COVID-19, an online survey and multivariate analysis are adopted to determine cyclists’ behavior.

The rest of the paper runs as follows. Section 2 presents the literature review and hypotheses development. Section 3 describes the methods and data collection. Section 4 discusses the results in comparison with previous research and presents the implications of the findings for theory and practice. The final section provides the conclusion, limitations of the study, and recommendations for future research.

## 2. Literature Review and Hypotheses Development

### 2.1. Bicycle Tourism

Bicycle tourism is a very popular recreational exercise in Taiwan, both before and during the COVID-19 pandemic. Compared to other forms of transportation, cycling allows people to remain at a safe distance between each other. The definition of bicycle tourism considers a bicycle trip away from an individual’s home region to be the main purpose of a trip, whether for active or passive participation in cycling [17]. Ritchie, Tkaczynski, and Faulks (2010) defined bicycle tourism as “tourism that involves watching or participating in a cycling event, or participating in independent or organized cycle touring” [18]. Bicycle tourism refers to cycling that involves spending at least one night away from a person’s home destination [19].

### 2.2. Motivation

Motivation is one of the main driving forces used to interpret the behavior of an individual and can be divided into push and pull factors. People are pushed by their internal forces and pulled by the external forces of destination attributes to participate in tourism activities [20]. Push factors are the forces pushing individuals from their home and to making a decision on tourism. Conversely, pull motivation cover factors that attract people toward a specific destination.

Cyclists’ motivations can vary depending on the individual and their destination [21], and they can also influence an individual’s decision and impel him/her to take an action [22,23]. A strong motivation could increase a person’s cycling ability and career [24]. The primary motivations of mountain biking tourists include challenges, excitement about social opportunities, and competitions [25]. The most important motives for sports participants are physical fitness and socializing, followed by knowledge and skills, relaxation, and personal development [26]. The motivations that push bicycle club members to become triathletes are adventure experiences, competence mastery, personal challenges, relaxation/escape, and social encounters [18]. For recreation and leisure pursuits, the pull motivations are the perceived attractiveness of a destination and its utility [27,28,29]. Havitz and Dimanche (1999) stated that involvement is an unobservable state of motivation, arousal, or interest towards a recreational activity [30]. It is evoked by a particular stimulus or situation that possesses some driving properties [5]. One of the most immediate impacts during the lockdown was a rise in bicycle use [31], since COVID-19 decreased people’s decision to participate in other tourism activities. Has the virus influenced tourists’ motivation? Is involvement just an unobservable motivation? This study explores the effect of COVID-19 in Taiwan with the following hypotheses:

**H1.** *COVID-19 cycling tourism positively and significantly affects cyclists’ motivation.*

**H2.** *COVID-19 cycling tourism positively and significantly affects cyclists’ involvement.*

### 2.3. Involvement

Involvement represents the perceived importance of engaging in an activity based on inherent motives, needs, and interests [30]. McIntyre (1989) introduced the concept of enduring involvement (EI) [32], which is the central life role of leisure activities for individuals and is conceptualized as three dimensions: attraction, self-expression, and centrality. Attraction represents an individual’s attachment and interest in an activity and the satisfaction received from it. Self-expression contains personal and social identities tied to the activity. Centrality is the extent of individuals’ lives surrounding the activity and their friends also being associated with the activity. In addition, psychological involvement is about an individual’s relevance to an activity or the degree to which an individual devotes him/herself to an activity and represents a cognitive linkage between oneself and the activity [33].

Getz and Anderson (2010) found that people who are highly involved in sports are primarily motivated by self-development in terms of athleticism and challenges, whereas social motives and extrinsic motives are less important [34]. Kyle, Absher, Norman, Hammitt, and Jodice (2007) developed a Modified Involvement Scale (MIS) that includes five dimensions: attraction, centrality, social bonding, identity expression, and identity affirmation [35]. They removed the social bonding items from centrality to construct a distinct dimension titled social bonding, distinguished the symbolic and expressive elements of EI, and established identity affirmation and identity expression through leisure experiences. Identity affirmation helps examine the degree of leisure opportunities from individual to individual. Identity expression helps examine the extent of leisure opportunities to express the individual to others.

Levels of sport involvement are relevant for reducing perceived constraints and for increasing motivation towards participation [36]. Motivation that initiates and maintains involvement in specific leisure activities could be understood through a specific outcome by an individual’s pursuit [37]. Personal relevance gives insight into recreationists’ motivation to engage and to continue to be involved in specific leisure behaviors [7]. Iwasaki and Havitz (2004) found that motivation strongly predicts involvement [38]. This leads to another hypothesis:

**H3.** *Motivation positively and significantly affects involvement.*

### 2.4. Behavior

People who are highly involved with a subject or an issue are likely to influence/change their future behaviors [39]. The ways involved in an individual’s leisure activities can yield insights on various aspects of that person’s behavior [40]. Ritchie (1998) noted that the involvement of cyclists can be put into two categories: inexperienced cyclists who are seeking competency, and mastery or experienced cyclists who are motivated by solitude and exploration [41]. Brey and Lehto (2007) stated that the more frequently people participate, the higher their level of behavioral involvement [42]. Wiley, Shaw, and Havitz (2000) suggested that people remain involved in activities because of valuable benefits for their overall health and wellbeing [43]. Leisure involvement, self-efficacy, and motivation are three widely studied psychological phenomena that can exert a strong influence on behavior [44]. Nowadays, scientists used neuromarketing tools to detect consumers brain’s mechanisms for understanding their behavior to optimize marketing strategies [45,46]. Therefore, we present the following three hypotheses:

**H4.** *Motivation positively and significantly affects behavior.*

**H5.** *Involvement positively and significantly affects behavior.*

**H6.** *Involvement has a mediation effect between motivation and behavior.*

## 3. Methods

### 3.1. Design and Analysis Methods

This study designed a questionnaire to investigate the motivation, involvement, and behavior of cyclists in Taiwan. The questionnaire for motivation is based on the attributes in the studies of Kulczyckia and Halpenny (2014) [21] and Kyle et al. (2006) [7]. Involvement was measured using a leisure involvement scale modified from Kyle et al. (2007) [35]. A five-point Likert-type scale is adopted (ranging from ‘strongly disagree’ to ‘strongly agree’) so that we can assess the face validity of the scale items and the general quality of the research design via pre-tests. Based on Fabrigar, Wegener, MacCallum, and Strahan (1999) [47], we conducted exploratory factor analysis (EFA) for a pilot study to provide a basis for specifying a confirmatory factor analysis (CFA) model in the subsequent study. EFA is used to extract the main items of motivation and involvement. The main factorial dimensions of motivation and involvement are designed/combined into structural equation modeling (SEM) to examine the relationships among motivation, involvement, COVID-19, and behavior (Hypotheses H1 to H5). Finally, the mediation effect of involvement between motivation and behavior is tested (H6). The research framework is depicted in Figure 1.

### 3.2. Data Collection

The survey was conducted on the Survey Cake platform from April to May in 2022. Only cyclists who have experienced bicycle tourism were selected, and the master list was provided by a travel agency in Taichung City. Based on Salamzadeh, Tajpour, Hosseini, and Salamzadeh (2022), when the population is more than 100,000 people, the assigned sample size is 385 [48]. To hit the 95% confidence level, 0.5 standard deviation, and 5% margin of error, the required sample size is 385. Cyclists who participated in bike tourism between 2020 and 2022 were asked to fill out the questionnaire. In total, 437 cyclists filled out the questionnaire, and 401 completed it, yielding a 91.76% response rate. Overall, 200 respondents were male and 201 were female. The age group in the survey with the greatest number is 186 cyclists for 21–30 years old. Table 1 lists the detailed sociodemographic characteristics of the respondents.

## 4. Results

### 4.1. Exploratory Factor Analysis for Motivation and Involvement

This study uses EFA to extract the major facets of motivation and involvement for cyclists. Bartlett’s test of sphericity and the KMO (Kaiser–Meyer–Olkin) technique are applied to examine the appropriateness of the factor analysis. The results show that the Bartlett tests for motivation and involvement are both significant. The KMO values of motivation and involvement are 0.925 and 0.939, respectively. The factors thus correlate and are appropriate for factor analysis.

To extract motivation and involvement items, the principal component method and varimax rotation are used. Initially there were 25 items for motivation; one item with a factor loading lower than 0.4 was dropped, and five factor dimensions were extracted. All 15 involvement items were extracted into two facets. Table 2 and Table 3 list the results of the factor analysis for motivation and involvement. As shown in the two tables, the eigen-values exceed 1 and can explain 64.74% and 65.22% of the variance.

For motivation, the first dimension is the ‘social and nature experience’, which accounts for 21.04% of the total variance with a reliability of 0.89. The other dimensions are ‘nature and culture learning’, ‘enhancing cycling skills’, ‘novel tasting’, and ‘enhancing fitness’ factors, which account for 15.97%, 12.03%, 8.62%, and 7.08% of the total variance and their reliabilities are 87.3%, 80.3%, 68.1%, and 63.8%, respectively. The 15 involvement items extracted two factors. One is ‘attraction and identity’, and the other is the ‘centrality and social’ aspect. The total variance is 36.65% and 28.58% and the reliability is 92.9% and 88.9%, respectively.

In addition to motivation and involvement, there are three items each for COVID-19 and behavior intention factors. The items for COVID-19 cycling tourism factor include: ’During the COVID-19 epidemic, cycling tourism allowed me to maintain a safe social distance from others’, ‘Because COVID-19 is an infectious disease, I got into cycling tourism’, and ‘Anxiety due to COVID-19 can be relieved with cycling tourism’. The items for behavior intention include: ’During the COVID-19 epidemic, I continue to participate in cycling tourism’, ‘I will revisit destinations through cycling tourism’, and ‘I would recommend others to take up cycling tourism’.

Based on the results of the EFA, this study follows the suggestions of Fabrigar et al. (1999) [47], putting the main factors of motivation and involvement into SEM to examine the relationship among motivation, involvement, and behavior intention.

### 4.2. Structural Model

Based on the main factorial dimensions from EFA, the model is tested by SEM. The assessment for the fit of the empirical data to the proposed models is examined using multiple criteria to evaluate different aspects of the adequacy of the postulated model. Following Kline’s (2005) [49] suggestions for model fit, this study performs four goodness-of-fit indices: χ^2^/df, comparative fit index (CFI) [50], root mean square residual (RMR) [51], and the root mean square error of approximation (RMSEA) [52]. The standards of indices for an adequate fit model include: χ^2^/df is less than 5, standard for fit indices of CFI value needs to be greater than 0.90, RMR is greater than 0.05, and RMSEA values should fall between 0.08 and 0.10 [51,53,54]. The results of the model fit reveal that χ^2^/df = 3.41, CFI = 0.81, RMR = 0.06, and RMSEA = 0.08. All indices of the structure model fit the data, except CFI, where the value 0.81 is close to 0.9. The structural equation model is summarized in Figure 2.

Common method variance (CMV) is the amount of spurious correlation between the variables. Inflating or deflating the findings between variables may lead to erroneous conclusions. Harman’s single-factor test is traditionally used to examine CMV in EFA [55]. Hult, Ketchen, Cavusgil, and Calantone (2006) suggested that using CFA via the chi-square difference test is more robust for a one-factor model versus a multifactor model [56]. If common method bias poses a serious threat to analysis and data interpretation, then a single latent factor can account for all manifest variables [57]. In this study the one-factor model yields χ^2^ = 12,995.7 with 990 degrees of freedom, compared with the measurement model with χ^2^ = 3179.5 and 933 degrees of freedom. The fit of the one-factor model is considerably worse than the measurement model. Thus, common method bias is not a serious threat in the study.

Table 4 lists the results of the main latent variables for the relationships between motivation, involvement, and behavior intention. The latent variables include motivation, involvement, behavior intention, and COVID-19. All of the items are significantly correlated. For cycling tourism, COVID-19 affects motivation and involvement positively and significantly, supporting both Hypotheses 1 and 2 (H1 and H2). The results validate the research of Havitz and Dimanche (1999) [30] and Iwasaki and Havitz (1998) [5] in that the COVID-19 pandemic is an unobservable motivation and also a particular stimulus or situation.

The results also show that motivation affects involvement positively and significantly, thus supporting Hypothesis 3 (H3). This is in line with the research of Kyle et al. (2006) [7] and Getz and Anderson (2010) [34]. However, motivation does not affect the behavior intention of cyclists and hence does not support Hypothesis 4 (H4). The result also means that during the COVID-19 pandemic, cyclists may consider the risk of the infectious disease, which then affects their behavior intention. The next section presents the mediation effect of involvement between motivation and behavior intention.

### 4.3. Mediation Effect of Involvement

Direct effect refers to an effect that is not mediated by other variables; indirect effect is an effect that is mediated by other variables [58]. This study explores the mediating role of involvement between motivation and behavior intention through indirect effect, which means cyclists’ motivation may not influence their behavior intention directly. The total effect of motivation impacts behavior intention indirectly and has to go through the involvement. As shown in Table 5, the direct effect value is 0.321 and indirect effect value is 0.514; together, the total effect is 0.834. Since the direct effect value is less than the indirect effect value and the path between motivation and behavior is not significant, it is considered to be partial mediation. The result support Hypothesis 6 (Figure 1, dotted line).

## 5. Discussion

This study examines the effect of COVID-19 on cyclists’ motivation and involvement. The results reveal that COVID-19 cycling tourism affects motivation positively and significantly. The finding that COVID-19 cycling tourism positively influences motivation is consistent with Weed’s (2020) research [31]. Since motivation is an antecedent of enduring involvement [5,6,7], the empirical results of this study indicate that motivation affects involvement positively and significantly.

The most important finding of the study is that, during the pandemic, motivation did not affect behavior intention directly. Cyclists make their decision through involvement when considering the risk of the COVID-19 pandemic. Involvement has a significant mediation effect between motivation and behavior intention. This is consistent with the research of Wiley et al. (2000) and Havitz et al. (2013) [43,44], who argued that people choose activities through involvement over time due to overall health and wellbeing. Leisure involvement, self-efficacy, and motivation are three important factors that exert a strong influence on behavior.

### Implication of the Research Findings for Theory and Practice

Motivation theoretically impacts involvement, which reveals that motivation is an antecedent of involvement. However, motivation does not impact behavior intention directly during COVID-19. Involvement acts as a mediation role between motivation and behavior intention. Therefore, motivation must transfer the effect to behavior intention by involvement indirectly. The results show that motivation is an antecedent of involvement, and involvement increases the motivation towards participation, thus supporting the research of Havitz and Dimanche (1999) [30] and Alexandris (2013) [36]. In practice, this study examines the effect of COVID-19 cycling tourism, showing its impacts on motivation and involvement significantly. COVID-19 cycling tourism can be seen as an unobservable effect on motivation. People will consider the influence of the virus to engage in various activities in greater detail. Their involvement in sports is primarily motivated by attraction, identity, centrality, and social factors.

## 6. Conclusions

After the outbreak of the COVID-19 pandemic, many countries set up travel restrictions, safe social distancing, and lockdowns. All of these measures have affected the way people go about their lives, including taking part in leisure activities and tourism. This study presents several findings on the subject in Taiwan. First, it confirms the phenomenon that Weed (2020) [31] found; that is, people have increased the use of bikes during the COVID-19 pandemic. COVID-19 cycling tourism has also positively affected cyclists’ motivation in Taiwan.

Second, COVID-19 cycling tourism has positively affected the involvement of cycling tourists. This denotes an unobservable state of motivation.

Third, involvement represents an individual’s engagement in an activity based on the inherent motives, needs, and interests [30]. Therefore, participants’ involvement in sports increased as their motivation toward participation increased [36]. The results support previous studies in that motivation is an antecedent of enduring involvement [5,6,7].

Fourth and finally, the main contribution of this study is that involvement has a mediation effect between motivation and behavior intention. Motivation did not affect behavior intention directly during the COVID-19 pandemic. The COVID-19 pandemic has certainly impacted the wellbeing of people. Since COVID-19 is an infectious disease, people may perceive involvement as a risk for their health and wellbeing. Although COVID-19 has had an impact on sports and tourism, some people have turned to bicycle tourism to avoid catching this infectious virus.

### Limitations and Future Research

First, sample selection was not random, because it is difficult to access and examine a larger population during the COVID-19 pandemic. In future research, the sample can be taken on-site, or a random face-to-face survey can be conducted.

Second, for obtaining more accurate results, neuroimaging techniques can be applied to detect consumers’ emotional and cognitive processes due to the COVID-19 pandemic. Detection machines and techniques may be more precise than surveys, but this is much more expensive than traditional social science methods.

Third, the non-market goods method can be used to estimate the monetary benefits to people under different scenarios and to explore the change in wellbeing for advanced cost and benefit analysis. The contingent behavior model (CBM) combines actual and intended behavior data, and the recreational benefits can be measured by calculating the consumer surplus between the demand function of actual trips and intended behavior trips [15]. Therefore, CBM may be one method to measure the recreation benefits under hypothetical scenarios in which the COVID-19 pandemic is imposed.

## Figures and Tables

**Figure 1 behavsci-12-00479-f001:**
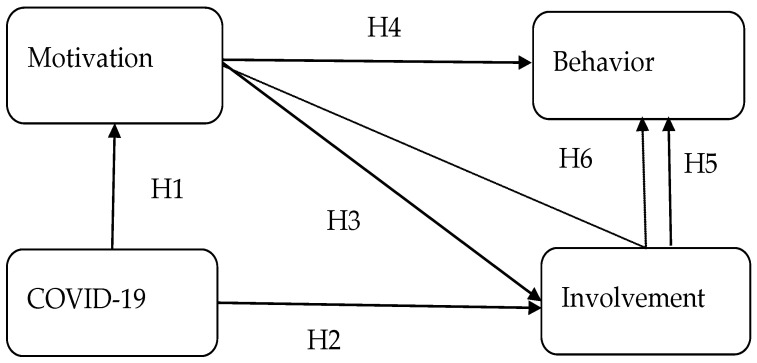
Research framework.

**Figure 2 behavsci-12-00479-f002:**
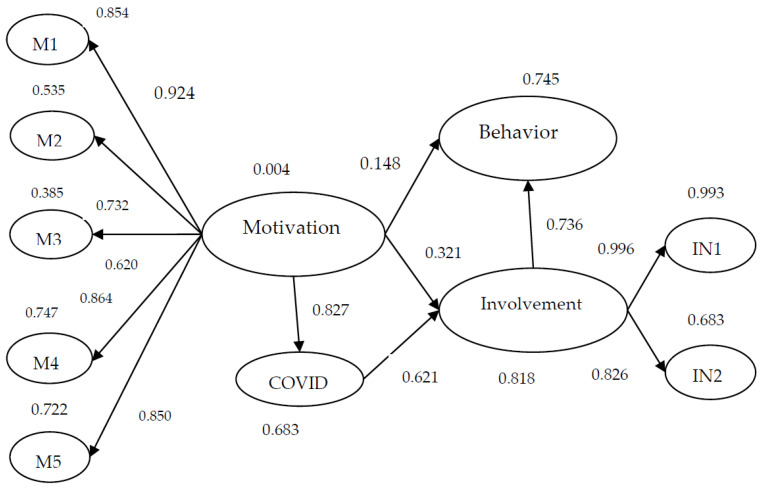
Result of the structural equation model.

**Table 1 behavsci-12-00479-t001:** Sociodemographic characteristics of respondents.

Characteristics	Frequency	Percent (%)
Gender		
Male	200	49.9
Female	201	50.1
Marital status		
Married	118	29.4
Single	277	69.1
Others	6	1.5
Age		
Under 20	42	10.5
21–30	186	46.4
31–40	82	20.4
41–50	46	11.5
Over 51	45	11.2
Education		
Junior high school	2	0.5
Senior high school	30	7.5
Undergraduate	282	70.3
Graduate school	87	21.7
Monthly Income		
Less than TWD 30,000	168	41.9
TWD 30,001–50,000	152	37.9
TWD 50,001–70,000	40	10.0
TWD 70,001–90,000	17	4.2
Over TWD 90,001	24	6.0

**Table 2 behavsci-12-00479-t002:** Factor analysis of motivation.

Item	Social and Nature Experience	Nature and Culture Learning	Enhancing Cycling Skills	Novel Tasting	Enhancing Fitness
To enjoy the scenery	0.777				
To escape my daily routine	0.688				
To be close to nature	0.675				
To share time with friends and family	0.674				
To do something with family and friends	0.643				
To enjoy time with people who do the same things as I do	0.634				
To seek out fun	0.618				
To experience a destination	0.530				
To experience new and different things	0.503				
To learn about the natural and cultural heritage		0.838			
To develop my knowledge of alpine history and culture		0.816			
To enjoy the culture and history of a mountain town		0.775			
To develop my knowledge of cycling		0.617			
To learn about nature		0.614			
To expand my cycling portfolio/record			0.816		
To develop my cycling skills and abilities			0.790		
To achieve my personal best in cycling/racing			0.767		
To engage in exercise			0.526		
To talk to new and varied people				0.699	
To engage in entertaining activities				0.698	
To do something different from what I normally do				0.464	
To consider COVID-19					−0.700
To have a stimulating and exciting experience					0.575
To experience physical challenge					0.567
Eigenvalue	9.478	1.877	1.832	1.300	1.051
Variance (Cumulative %)	21.042	37.012	49.037	57.655	64.739
Reliability (Cronbach’s α, %)	88.7	87.6	80.3	68.1	63.8

**Table 3 behavsci-12-00479-t003:** Factor analysis of involvement.

Item	Attraction and Identity	Centrality and Social
I identify with people and the image associated with cycling	0.808	
When I am cycling, I do not have to be concerned with the way I look	0.799	
You can tell a lot about a person by seeing them camping	0.769	
When I participate in camping, others see me the way I want them to see me	0.744	
Cycling is one of the most enjoyable things I do	0.736	
Cycling is one of the most satisfying things I do	0.715	
Participating in camping says a lot about who I am	0.707	
Cycling is very important to me	0.607	
When I participate in cycling, I can really be myself	0.510	
Most of my friends are in some way connected with cycling		0.841
Cycling occupies a central role in my life		0.770
I find a lot of my life is organized around cycling		0.746
To change my preference for cycling to another recreation activity would require major rethinking		0.706
I enjoy discussing cycling with my friends		0.681
Participating in camping provides me with opportunity to be with friends		0.634
Eigenvalue	8.438	1.346
Variance (Cumulative, %)	36.646	65.221
Reliability (Cronbach α, %)	92.9	88.9

**Table 4 behavsci-12-00479-t004:** Main latent variables’ structure path.

	Path		Proposed Direction	Estimates	t Value	Result
COVID-19	→	Motivation	+	0.827 ***	8.74	Supports H1
COVID-19	→	Involvement	+	0.621 ***	4.30	Supports H2
Motivation	→	Involvement	+	0.321 ***	2.71	Supports H3
Motivation	→	Behavior	+	0.148	1.70	Does not support H4
Involvement	→	Behavior	+	0.736 ***	7.77	Supports H5

Note: *** represents significance at *p* < 0.01.

**Table 5 behavsci-12-00479-t005:** Mediation effect of involvement.

	Direct Effect	Indirect Effect	Total Effect
Motivation	0.321	0.514	0.834
Behavior intention	0.736	-	0.736

## Data Availability

The data are available by correspondence with the author.

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
