# Peer review of "The Impact of COVID-19 on Motivation, Involvement, and Behavior of Cyclists in Taiwan"

_behavsci, 2022, doi:10.3390/bs12120479_

Round 1

Reviewer 1 Report

Thank you for writing this timely review. While the topic is indeed of great value to both academia and the practitioner communities, there remain few issues that ought to be addressed before this paper can be accepted.

I suggest the following revisions to strengthen the paper further:

1.     To begin with, this paper needs English editing. In its present form, there is some not readable at various parts.

2.     The authors should explicitly state the novel contribution of this work and its similarities and differences with their previous publications.

3.     The authors need to clearly articulate the academic as well as practical implications of this study in a separate  section which can named the theoretical and practical implication of this study. I suggest reference, which can be benefit for this issue  doi.org/10.1080/23311975.2021.1978620;

4.     I think that in the “Motivation, involvement, and behavior” section need some development. I suggest updated reference, which can be beneficial for improving the introduction, i.e., doi.org/10.31117/neuroscirn.v4i3.79

5.     What criteria does the author base on to select the experimental sample (number of questionnaires sent out and received)? As stated in the article, the author has chosen 437 copies and obtained a valid piece of 401. What is the statistical significance of 401 questionnaires for this study?

6.     The authors did not refer to the gender and age of participants in the questionnaire, the authors should explained this issue? How many participants and age of males and female in table?

7.     The authors need to clarify the limitation and future directions in a separate section, I would suggest preparing a separate section for the limitations and future directions after discussion & Conclusions section (i.e., 7. limitations and future research). For that issue, I would like to suggest a reference which can help the authors doi.org/10.3991/ijim.v16i13.30605; doi.org/10.47743/saeb-2022-0020; doi.org/10.3991/ijoe.v17i10.25243

8.     The authors need to clearly articulate the key implications at the end of the 'Introduction' section. I suggest article which can be benefits to improve that issue "neuroimaging techniques in advertising research: main applications, development, and brain regions and processes".

9.     For readers to quickly catch your contribution, it would be better to highlight major difficulties and challenges and your original achievements to overcome them in a clearer way in the abstract and introduction.

10.  How could/should futures studies improve the model?

If these revisions can be made in the manuscript, I believe that this study can be accepted for publication.

I wish the authors all the very best with this study.

Author Response

Dear reviewers 1,

We appreciate your helpful comments, guidance to revise the manuscript, and provides useful information to us. We hope the following explanations could answer all your questions.

  1. The revised manuscript has been checked by an editor who has a British English PHD.

  1. We have added the novel contribution of this work, and described differences with their previous publications in the second paragraph of ‘Introduction” section as follows,

Many studies on COVID-19 have focused on motivation (Bhatta, Gautam, & Tanaka, 2022) [10], motivation and intention (Sageng, Ting, Chang, Leong, & Ting, 2021) [11], motivation and constraint (Humagain & Singleton, 2021) [12], or behavior and intention (Aydin, Kuşakcı & Deveci, 2022; Fan, Lu, Qiu, & Xiao, 2022) [13-14] etc. So far, none has discussed tourists’ concern over COVID-19 as well as the relationships among their motivation, involvement, and intention. This study is the first to present the depth and breadth of COVID-19 influences upon tourism (please see from line 58 to 64).

  1. We follow the reviewer’s suggestion and have added a subsection, ‘5.1. Implication of the research findings for theory and practice’.

Theoretically, motivation impacts involvement, which reveals that motivation is an antecedent of involvement. However, motivation does not impact behavior intention directly. Involvement acts as a mediation role between motivation and behavior intention. Therefore, motivation must transfer the effect to behavior intention by involvement indirectly. The results show that motivation is an antecedent of involvement, and involvement increases motivation towards participation, thus supporting the research of Havitz and Dimanche (1999) [29] and Alexandris (2013) [35]. In practice, this study examines the effect of COVID-19 cycling tourism, showing its impacts on motivation and involvement significantly. COVID-19 cycling tourism can be seen as an unobservable effect on motivation. People will consider the influence of the virus to engage in the activities in greater detail. Their involvement in sports are primarily motivated by attraction, identity, centrality, and social factors (please see from line 368 to 379).

  1. We have rewritten the subsection “2. Literature Review and Hypotheses Development” and add more updated reference for better understanding. Now, it has divided into 2.1 Bike tourism, 2.2. Motivation, 2.3 Involvement, and 2.4 Behavior sections.

  1. The master list was provided by the travel agency in Taichung City, and only cyclists who have experienced bicycle tourism was selected. Based on Salamzadeh, Tajpour, Hosseini, and Salamzadeh (2022), to hit the 95% confidence level, 0.5 standard deviation, and 5% margin of error, the required sample size is 385. (please see from line 198 to 202, in the subsection 3.2. Data collection).

  1. The demographics information was added in Table 1. Two hundred respondents were male and 201 were female. The age group in the survey with the greatest number is 186 cyclists for 21-30 years old (please see from line 205 to 207, in subsection 3.2. Data collection).

  1. We added the subsection “6.1. limitations and future research” in 6. Conclusions section as follows,

6.1. Limitations and future research

The limitation of this study is that the sample selection was not random, because it is difficult to access and conduct larger population during COVID-19 pandemic. In future research, the sample can be taken on-site, or a random face-to-face survey can be made. As previous research focused on perception among the factors, future research can adopt the non-market goods method to estimate the monetary benefits to people under different scenarios and to explore the change of well-being for advance cost and benefit analysis. The Contingent Behavior Model (CBM) combines actual and intended behavior data, the recreational benefits can be measured by calculating the consumer surplus between the demand function of actual trips and intended behavior trips [15]. Therefore, CBM may be one method to measure the recreation benefits under hypothetical scenarios that the COVID-19 pandemic is imposed (please see from line 404 to 414).

  1. Base on Alsharif, A. H. et al. (2021), we added the implications and described the development of this study at the introduction section (please see from line 71 to 80).

To understand the COVID-19 impacts on cycling tourism, the research first stems the mo-tivation of cyclists. Then, analyzing the involvement of cyclists via their inherent motives, needs, and interests in regards to the COVID-19. Finally, it explores the response behavior of cyclists to motivation and involvement under the COVID-19 pandemic.

The rest of the paper arranges as follows. Section 2 presents the literature review and hypotheses development. Section 3 describes the methods and data collection. Section 4 discusses the results with previous research and offers implications of the findings for theory and practice. The final section gives a conclusion, limitation, and future research direction.

  1. For readers to quickly understand our original achievements, we have restructured the abstract. Also, we highlighted major difficulties and challenges and provided possible ways to overcome these issues at 6.1. Limitations and future research.

The limitation of this study is that the sample selection was not random, because it is difficult to access and conduct larger population during COVID-19 pandemic. In future research, the sample can be taken on-site, or a random face-to-face survey can be made (please see from line 404 to 407).

  1. In the future, researchers may adopt Contingent Behavior Model (CBM), which can measure the recreational benefits changes under various hypothesis scenarios, such as the COVID-19 pandemic.

The Contingent Behavior Model (CBM) combines actual and intended behavior data, the recreational benefits can be measured by calculating the consumer surplus between the demand function of actual trips and intended behavior trips [15]. Therefore, CBM may be one method to measure the recreation benefits under hypothetical scenarios that the COVID-19 pandemic is imposed (please see from line 410 to 414).

Reviewer 2 Report

Dear authors

It was my pleasure to review your manuscript entitled “The impact of covid-19 on motivation, involvement and behavior of cyclist in Taiwan” and advise you to prosper your current research project. In my view, your topic has touched on a critical issue in a fascinating context. However, there are many spaces to be improved in terms of argumentation, theoretical background, research method, and findings. I hope my below comments would help you develop your work into groundbreaking research in your domain.

Positioning, purpose, introduction and research gap.

The abstract should include the context and purpose, research method, findings, and results. The abstract should also indicate the innovation of the work.

The introduction should clearly illustrate (1) what we know (the key theoretical perspectives and empirical findings) and what we do not know (major, unaddressed puzzle, controversy, or paradox does the study address, or why it needs to be addressed and why this matters) and (2) what we will learn from the study, and how the study fundamentally changes, challenges, or advances scholars’ understanding. Much sharper problematization is required so that the introduction draws the reader into the paper. The introduction, therefore, needs to do a better job of setting the stage for the articulation of the theoretical contributions of the study.

Please provide more information on the context of your study. I would like to better understand the context of your empirical work

.•          Remedy:  Explain 2 lines about the importance and contribution at the introduction end of the article.

           Theoretical literature has not been considered and reviewed. It’s better to observe the connection between the contents. Try to explain everything except the topics in order to establish the necessary coherence.

           Remedy: Please provide more information on the context of your study. I would like to understand the context of your empirical work better: what about the impact of covid-19 on motivation, involvement and behavior of cyclist? And what can be learned from selecting observations in this area?

Method.

           Insufficient transparency. The authors need to provide and explain more details on their method, including their sampling, data gathering and data analysis.

           What basis is the statistical population estimated?

           What were the criteria of the respondents?

           How the validity and reliability of the questionnaire were measured before sending the questionnaires.

Discussion.

           In this section, examine the hypotheses separately and compare them with the literature section.

Conclusion.

The conclusion shows the final results of your research (you need a conclusion for your research).

Please clarify what are the theoretical and practical contributions of your research.

The authors need to draw substantive conclusions from their results and suggest, develop recommendations for further research.

** Considering that the sources used in the article are not up-to-date, the authors should use at least 6 up-to-date articles from 2021-2022.

           Using the following reference could be beneficial as these add more evidence to the literature review section:

Salamzadeh, A., Tajpour, M., Hosseini, E., & Salamzadeh, Y. (2022). Geotourism and Destination Brand Selection: Does Social Media Matter? In Economics and Management of Geotourism (pp. 105-124). Springer.

Best of luck with the further development of the paper.

Author Response

Dear reviewer 2,

We appreciate your helpful comments for revising the manuscript. We hope the following explanations could answer all your questions.

  1. We have added the innovation in abstract.

So far, none has discussed tourists’ concern over COVID-19 as well as the relationships among their motivation, involvement, and behavior intention. This study tries to explore the relationship (please see from line 15 to 16).

During the COVID-19 period, the challenge has been to perform face-to-face investigation on-site. This study thus employs an online survey for data analysis, fills the gap in the literature to discuss the involvement of tourists concerning COVID-19, and presents the depth and breadth of COVID-19 effects upon tourism (please see from line 24 to 27).

  1. We have added the description the contribution of this study in introduction (please see from line 61 to 64).

So far, none has discussed tourists’ concern over COVID-19 as well as the relationships among their motivation, involvement, and behavior intention. This study is the first to present the depth and breadth of COVID-19 influences upon tourism.

  1. We have added to review theoretical literature in introduction and Literature.

Many studies on COVID-19 have focused on motivation (Bhatta, Gautam, & Tanaka, 2022) [10], motivation and intention (Sageng, Ting, Chang, Leong, & Ting, 2021) [11], mo-tivation and constraint (Humagain & Singleton, 2021) [12], or behavior and intention (Ay-din, Kuşakcı & Deveci, 2022; Fan, Lu, Qiu, & Xiao, 2022) [13-14] etc. (please see from line 57 to 61)

Ritchie, Tkaczynski, and Faulks (2010) defined bicycle tourism as “tourism that involves watching or participating in a cycling event, or participating in independent or organized cycle touring” [17]. Or, bicycle tourism refers to cycling that spends at least one night from a person’s home destination (Han, Meng, & Kim, 2017) [18]. (please see from line 88 to 91)

Motivation is one of the main driving forces used to interpret the behavior of an indi-vidual and can be divided into push and pull factors. People are pushed by their internal forces and pulled by the external forces of destination attributes to participate in tourism activities (Crompton, 1979) [19]. Push factors are the forces pushing individuals from their home and to make a decision on tourism. Conversely, pull motivation cover factors that attract people toward a specific destination. (please see from line 94 to 99)

McIntyre (1989) introduced the concept of Enduring Involvement (EI) [31]. This is central life role of leisure activities for individuals, and is conceptualized as three dimensions: attraction, self-expression, and centrality. Attraction represents an individual’s attachment and interest in an activity and the satisfaction receives from it. Self-expression contains personal and social identities tied to the activity. Centrality is the extent of individuals’ lives surrounding the activity and their friends also being associated with the activity. (please see from line 127 to 133)

  1. We added to describe the conception development and empirical work of this study as follows (please see from line 71 to 75),

To understand the COVID-19 impacts on cycling tourism, the research first stems the motivation of cyclists. Then, analyzing the involvement of cyclists via their inherent motives, needs, and interests in regards to the COVID-19. Finally, it explores the response behavior of cyclists to motivation and involvement under the COVID-19 pandemic.

  1. We have added to describe the criteria of sampling in ‘3.2. data collection’ and ‘6.1. Limitations and future research” as below. Prior to data collection.

We assessed the face validity of the scale items and the general quality of the research design via pretests involving three academics. (please see from line 186 to 187)

The survey was conducted on the Survey Cake platform from April to May in 2022. Only cyclists who have experienced bicycle tourism was selected, and the master list was provided by the travel agency in Taichung City. Based on Salamzadeh, Tajpour, Hosseini, and Salamzadeh (2022), when the population is more than 100,000 people, the assigned sample size is 385 [45]. To hit the 95% confidence level, 0.5 standard deviation, and 5% margin of error, the required sample size is 385. (please see from line 198 to 203)

The limitation of this study is that the sample selection was not random, because it is difficult to access and conduct larger population during COVID-19 pandemic. In future research, the sample can be taken on-site, or a random face-to-face survey can be made. (please see from line 404 to 407)

  1. We added the subsection “5.1. Implication of the research findings for theory and practice” in discussion section. (please see from line 368 to 379)

Theoretically, motivation impacts involvement, which reveals that motivation is an antecedent of involvement. However, motivation does not impact behavior intention directly. Involvement acts as a mediation role between motivation and behavior intention. Therefore, motivation must transfer the effect to behavior intention by involvement indirectly. The results show that motivation is an antecedent of involvement, and involvement increases motivation towards participation, thus supporting the research of Havitz and Dimanche (1999) [29] and Alexandris (2013) [35]. In practice, this study examines the effect of COVID-19 cycling tourism, showing its impacts on motivation and involvement significantly. COVID-19 cycling tourism can be seen as an unobservable effect on motivation. People will consider the influence of the virus to engage in the activities in greater detail. Their involvement in sports are primarily motivated by attraction, identity, centrality, and social factors.

  1. We added the subsection “6.1. limitations and future research” in 6. Conclusions section as follows, (please see from line 404 to 414)

The limitation of this study is that the sample selection was not random, because it is difficult to access and conduct larger population during COVID-19 pandemic. In future research, the sample can be taken on-site, or a random face-to-face survey can be made. As previous research focused on perception among the factors, future research can adopt the non-market goods method to estimate the monetary benefits to people under different scenarios and to explore the change of well-being for advance cost and benefit analysis. The Contingent Behavior Model (CBM) combines actual and intended behavior data, the recreational benefits can be measured by calculating the consumer surplus between the demand function of actual trips and intended behavior trips [15]. Therefore, CBM may be one method to measure the recreation benefits under hypothetical scenarios that the COVID-19 pandemic is imposed.

  1. We have added 6 literatures in reference as follows,

Salamzadeh, A.; Tajpour, M.; Hosseini, E.,; Salamzadeh, Y. Geotourism and Destination Brand Selection: Does Social Media Matter? In Economics and Management of Geotourism (pp. 105-124). Springer, 2022.

Bhatta, K.; Gautam, P.; Tanaka, T. Travel Motivation during COVID-19: A Case from Nepal. Sustainability, 2022, 14, 7165.

Sageng, C. W.; Ting, H.; Chang, H. H.; Leong, C. M.; Ting, H. B. Motivation factors driving travel intention in the controlled pandemic context: Perspectives from Malaysian and Taiwanese travelers. Asian J. Bus. Res. 2021, 11(3), 92-112.

Humagain, P.; Singleton, P. A. Exploring tourists’ motivations, constraints, and negotiations regarding outdoor recreation trips during COVID-19 through a focus group study. J. Outdoor Recreat. Tour. 2021, 36, 100447.Aydin, N.; Kuşakcı A. O.; Deveci, M. The impacts of COVID-19 on travel behavior and initial perception of public transport measures in Istanbul. Decis. Anal. 2022, 2, 100029.

Fan, X.; Lu, J.; Qiu, M.; Xiao, X. Changes in travel behaviors and intentions during the COVID-19 pandemic and recovery period: A case study of China. J. Outdoor Recreat. Tour. 2022, online, doi.org/10.1016/j.jort.2022.100522.

Reviewer 3 Report

Dear Authors,

Comment 1: Abstract: The methods section should describe data collection methods and data analysis methods.

Comment 2: Introduction: I suggest deleting the first paragraph, which is not very relevant to the topic of the paper.

Comment 3: In the introduction, the authors briefly introduced the background and significance of the research. I consider it necessary to highlight the connection between the research objectives and the research gaps. The novelty of the study and the highlighting of the main conclusions should also be emphasized. Besides, a description of the structure of the paper should be added.

Comment 4: Line 54: “Materials and Methods” should be changed to “Literature Review and Hypotheses Development”.

Comment 5: Line 123: 2.3. Method and sample should stand alone as Chapter 3, Methods. In addition, Chapter 3 should be divided into several parts: data collection, measurement, and data analysis. The current content of 2.3 is obviously insufficient and incomplete, please describe more details.

Comment 6: Line 125: How is COVID-19 measured, please describe.

Comment 7: Figure 1 should be redrawn, keeping symmetry and aesthetics.

Comment 8: Results: The issue of CMV should be addressed.

Comment 9: I do not agree with the authors directly using the results of AMOS as Figure 2. Figure 2 should be drawn by the authors, indicating the factor loading and the square of R.

Comment 10: The content of the discussion part is obviously insufficient and should be divided into two sections: theoretical contribution, and practical implications.

Comment 11: Limitations should be more concise and moved to the Conclusion section.

Author Response

Dear reviewer 3,

We really appreciate your valuable comments. Thanks for your comments, we are able to improve the quality of the manuscript. The manuscript has been revised following your suggestions. We hope the following answers may response all of your queries.

  1. We added the description of analysis methods and data collection method in abstract as follow.

First, exploratory factor analysis is used to extract main factors of motivation, involvement and behavior items. Second, the structure equation model is constructed to examine the relationship between motivation, involvement and behavior intention (please see from line 16 to 19). During the COVID-19 period, the challenge has been to perform face-to-face investigation on-site. This study thus employs an online survey for data analysis, fills the gap in the literature to discuss the involvement of tourists concerning COVID-19, and presents the depth and breadth of COVID-19 effects upon tourism (please see from line 24 to 27).

  1. We deleted the first paragraph in introduction.

  1. We added to describe the research gaps, novelty, connection, and paper structure in introduction section as below.

Many studies on COVID-19 have focused on motivation (Bhatta, Gautam, & Tanaka, 2022) [10], motivation and intention (Sageng, Ting, Chang, Leong, & Ting, 2021) [11], motivation and constraint (Humagain & Singleton, 2021) [12], or behavior and intention (Aydin, Kuşakcı & Deveci, 2022; Fan, Lu, Qiu, & Xiao, 2022) [13-14] etc. So far, none has discussed tourists’ concern over COVID-19 as well as the relationships among their motivation, involvement, and intention. This study is the first to present the depth and breadth of COVID-19 influences upon tourism (please see from line 57 to 63).

To understand the COVID-19 impacts on cycling tourism, the research first stems the motivation of cyclists. Then, analyzing the involvement of cyclists via their inherent motives, needs, and interests in regards to the COVID-19. Finally, it explores the response behavior of cyclists to motivation and involvement under the COVID-19 pandemic (please see from line 71 to 75).

The rest of the paper arranges as follows. Section 2 presents the literature review and hypotheses development. Section 3 describes the analysis methods and data collection. Section 4 discusses the results with previous research and offers implications of the findings for theory and practice. The final section gives a conclusion, limitation, and future research direction (please see from line 76 to 80).

  1. We replaced “Materials and Methods” with “Literature Review and Hypotheses Development”.

  1. The subsection “2.3. Methods and Sample” has revised to “3. Methods”, and divided into “3.1. Design and analysis methods” and “3.2. Data collection” subsection.

  1. COVID-19 is measured in 3 items and used EFA to extract to one factor. A five-point Likert-type scale is adopted (ranging from `strongly disagree' to `strongly agree').

The items for COVID-19 cycling tourism factor include: ’During the COVID-19 epidemic, cycling tourism allowed me to maintain a safe social distance from others’, ‘Because COVID-19 is an infectious disease, I got into cycling tourism’, and ‘Anxiety due to COVID-19 can be relieved with cycling tourism’. The items for behavior intention include:  ’During the COVID-19 epidemic, I continue to participate in cycling tourism’, ‘I will revisit destinations through cycling tourism’, and ‘I would recommend others to take up cycling tourism’ (please see from line 236 to 243).

  1. We have redrawn Figure 1.

  1. We added to address the common method variance in subsection of Structural Model as below.

Common Method Variance (CMV) is the amount of spurious correlation between variables. Inflating or deflating findings between variables may lead to erroneous conclu-sions. Harman’s single-factor test is traditionally used to examine CMV in EFA (Craighead, Ketchen, Dunn, & Hult, 2011) [52]. Hult, Ketchen, Cavusgil, and Calantone (2006) suggested that using CFA via the chi-square difference test is more robust for a one-factor model versus a multifactor model [53]. If common method bias poses a serious threat to the analysis and data interpretation, then a single latent factor can account for all manifest variables (Podsakoff & Organ, 1986) [54]. In this study the one-factor model yield χ2 = 12995.7 with 990 degrees of freedom, compared with the measurement model with χ2 = 3179.5 and 933 degrees of freedom. The fit of one-factor model is considerably worse than the measurement model. Thus, common method bias is not a serious threat in the study (please see from line 300 to 311).

  1. We have redrawn Figure 2.

  1. We added the subsection “5.1. Implication of the research findings for theory and practice” in Discussion section.

Theoretically, motivation impacts involvement, which reveals that motivation is an antecedent of involvement. However, motivation does not impact behavior intention directly. Involvement acts as a mediation role between motivation and behavior intention. Therefore, motivation must transfer the effect to behavior intention by involvement indirectly. The results show that motivation is an antecedent of involvement, and involvement increases motivation towards participation, thus supporting the research of Havitz and Dimanche (1999) [29] and Alexandris (2013) [35]. In practice, this study examines the effect of COVID-19 cycling tourism, showing its impacts on motivation and involvement significantly. COVID-19 cycling tourism can be seen as an unobservable effect on motivation. People will consider the influence of the virus to engage in the activities in greater detail. Their involvement in sports are primarily motivated by attraction, identity, centrality, and social factors (please see from line 368 to 379).

  1. We added the subsection “1. limitations and future research” in Conclusion section.

The limitation of this study is that the sample selection was not random, because it is difficult to access and conduct larger population during COVID-19 pandemic. In future research, the sample can be taken on-site, or a random face-to-face survey can be made. As previous research focused on perception among the factors, future research can adopt the non-market goods method to estimate the monetary benefits to people under different scenarios and to explore the change of well-being for advance cost and benefit analysis. The Contingent Behavior Model (CBM) combines actual and intended behavior data, the recreational benefits can be measured by calculating the consumer surplus between the demand function of actual trips and intended behavior trips [15]. Therefore, CBM may be one method to measure the recreation benefits under hypothetical scenarios that the COVID-19 pandemic is imposed (please see from line 404 to 414).

Reviewer 4 Report

Dear colleagues

The submitted article reflects a very thorough approach the authors stick to while exploring the topic. The research is perfectly done and soundly based on the relevant scientific literature. All the six hypotheses suggested by the authors are supported with a thoroughly done analysis.

Yet, there are some misprints in the text that should draw the authors' attention.

The list of misprints:

  1. Line 27-28:  according to the World Health Organization’s (WHO) report on 10 August 2022.
  2. Line 30: The COVID-19 has positively and significantly influence on  (positively and significantly influenced / a positive and significant influence on) our life
  3. Line 42: had also negative impacted on   (had also negatively impacted / had also a negative impact on) many countries’ economy
  4. Line 47: on people devoting themself  (-selves) to
  5. Line 49: how the COVID-19 pandemic impacts on motivation and behavior of tourists? .
  6. Lines 50-51: This study will examine f evidence from Taiwan.

Lines 56-57: Bicycle tourism is one (a) popular sport in Taiwan, before and during the period of pandemic because compare to  (compared with / comparing with) other transportation,

  1. Line 58:  In according According to
  2. Line 127: Involvement were  (was) measured
  3. Line 133: were designed/combined?
  4. cyclists who have participated in bike tourism between 2020 and 2022
  5. Line 276: affected the way people live their life (lives), including their leisure activities and tourism
  6. Line 277: study has serval findings in Taiwan.

Kind regards,

Author Response

Dear reviewer 4,

We really appreciate your valuable and kind comments. Thanks for your comments, we are able to improve the quality of the manuscript. The manuscript has been revised following your suggestions. Thank you.

Reviewer 5 Report

This paper studies the influence of COVID-19 on cyclists' motivation, participation and behavior. The structural equation models test it, and some interesting conclusions are obtained. But there are some problems with the research idea and process.

1. "2.1 Bicycle tourism and motivation" and "2.2 Promotion, involvement and behavior" in the "2 Materials and Methods" section contain a large number of literature reviews. The general title of these two parts should be "Literature review and hypothesis development."

2 It is pointed out that COVID-19 pandemics include "During COVID-19 empirical, cycling tourism allowed me to maintain a safe social distance from others", "Because COVID-19 is an effective disease, I got into cycling tourism", and "Happiness due to COVID-19 can be released with cycling tourism". These questions show the relationship between the COVID-19 pandemic and cycling tourism, so they cannot represent the "COVID-19 pandemic" variable.

3 In line 259, the author pointed out that "The finding that the COVID-19 pandemic increased the use of bikes is consistent with Weed (2020)" However, in line 277, the author points out that "this study confirmed the phenomenon that Weed (2020) found. So what are the differences between the conclusions of this part and those of the cited literature?

4 In line 260, the author points out that "Since involvement is an unobservable state of motivation," since "involvement" is a kind of "motivation," why are "involvement" and "motivation" also two different variables in the text?

Author Response

Dear reviewer 5,

We really appreciate your valuable comments. Thanks for your comments, we are able to improve the quality of the manuscript. The manuscript has been revised following your suggestions. We hope the responses below answer all of your queries.

  1. We replaced“Materials and Methods”with“Literature Review and Hypotheses Development”.

  1. We changed “COVID-19 pandemic” to “COVID-19 cycling tourism”.

  1. The original sentence of line 259,” The finding that the COVID-19 pandemic increased the use of bikes is inconsistent with Weed (2020)”. We revised it to “The finding that COVID-19 cycling tourism positively influences motivation is consistent with Weed’s (2020) research“(see in line 350 of revision text).

  1. We have corrected the original sentence in line 260 to “Since motivation is an antecedent of enduring involvement (Iwasaki & Havitz, 1998; Funk, Ridinger & Moorman, 2004; Kyle, Absher, Hammitt, & Cavin, 2006) [5-7].” (see in line 354 of revision text).

Round 2

Reviewer 1 Report

Thank you for writing this timely review. While the topic is indeed of great value to both academia and the practitioner communities, there remain few issues that ought to be addressed before this paper can be accepted.

I suggest the following revisions to strengthen the paper further:

1.     To begin with, this paper needs English editing. In its present form, there is some not readable at various parts.

2.     The majority of references are outdated and need updated.

3.     I think that in the “Motivation, involvement, and behavior” section need some development and recent references because the references that used in that section are outdated. I suggest updated references, which can be beneficial for improving the motivation, involvement, and behavior subsections, i.e., doi.org/10.31117/neuroscirn.v4i3.79; https://doi.org/10.3991/ijoe.v18i08.31959

4.     The authors need to clearly articulate the key implications at the end of the 'Introduction' section. I suggest article which can be benefits to improve that issue, i.e., neuroimaging techniques in advertising research: main applications, development, and brain regions and processes.

5.     How could/should futures studies improve the model?

If these revisions can be made in the manuscript, I believe that this study can be accepted for publication.

I wish the authors all the very best with this study.

Author Response

Dear reviewers 1,

We appreciate your helpful comments and guidance to revise the manuscript, again. We hope the following explanations could answer all your questions.

  1. The revised manuscript has been checked again by another editor who is a native English speaker and has a PhD.

  1. We have added 3 Neuroimaging techniques literatures in the manuscript and reference as follows,

Alsharif, A.H.; Salleh,N. Z. M.; Baharun, R.; Hashem E, A.R.; Mansor, A. A.; Ali, J.; Abbas, A.F. Neuroimaging techniques in advertising research: Main applications, development, and brain regions and processes. Sustainability 2021, 13, 6488.

Alsharif, A.H.; Salleh,N.Z.M.; Baharun, R. Neurosci. Neuromarketing: Marketing research in the new millennium. Neurosci. Res. Notes, 2021, 4 (3), 27-35.

Alsharif, A. H.; Salleh,N. Z. M.; Baharun, R.; Hashem E, A.R. Neuromarketing research in the last five years: a bibliometric analysis. Cogent Bus. Manag. 2021, 8, 1978620.

  1. We follow the reviewer’s suggestions, and add more literature in “4. Behavior.

Nowadays, scientists used neuromarketing tools to detect consumers brain's mechanisms for understanding their behavior to optimize marketing strategies (Alsharif, Salleh, Baharun, 2021; Alsharif, Salleh, Baharun, Hashem, 2021) [57-58].

  1. We also follow the reviewer’s suggestion to add the description in “ Introduction” as below.

Interdisciplinary research, such as neuroimaging techniques, can be applied in social science (Alsharif, Salleh, Baharun, Hashem, Mansor, Ali, Abbas, 2021) [56], but to prevent the infection of COVID 19, online survey and multivariate analysis are adopted to decompose cyclists’ behavior.

  1. For getting more accurate results in the future, neuroimaging techniques can be applied to detect consumers’ emotional and cognitive processes due to the COVID-19 pandemic. Detection machines and techniques may be more precise than surveys, but this is much more expensive than traditional social science methods.

Sincerely,

Pr. Dr. Huang

Reviewer 2 Report

Dear author(s)

Hope you are doing well. According to the review of this article, the corrections have been made.

Good luck

Author Response

Dear reviewers 2,

We appreciate your comments, again.

The revised manuscript has been checked again by another editor who is a native English speaker and has a PhD.

Sincerely,

Pr. Dr. Huang

Reviewer 3 Report

Thank you for your revision.

Author Response

Dear reviewers 3,

We appreciate your comments, again.

The revised manuscript has been checked again by another editor who is a native English speaker and has a PhD.

Sincerely,

Pr. Dr. Huang